# Towards Boosting the Open-Domain Chatbot with Human Feedback

## Abstract

Many open-domain dialogue models pre-trained with social media comments can generate coherent replies but have difficulties producing engaging responses. This phenomenon might mainly result from the deficiency of annotated human-human conversations and the misalignment with human preference. In this paper, we propose a novel and efficient framework Diamante to boost the open-domain chatbot, where two kinds of human feedback (including explicit demonstration and implicit preference) are collected and leveraged. By asking annotators to select or amend the model-generated candidate responses, Diamante efficiently collects the human demonstrated responses and constructs a Chinese chit-chat dataset. To enhance the alignment with human preference, Diamante leverages the implicit preference in the data collection process and introduces the generation-evaluation joint training. Comprehensive experiments indicate that the Diamante dataset and joint training paradigm can significantly boost the performance of pre-trained dialogue models. The overall engagingness of the previous state-of-the-art model has been improved remarkably by 50% in Chinese open-domain conversations.

## 1 Introduction

In recent years, the self-supervised pre-training based on tremendous unlabeled data has brought great success for many natural language processing tasks (Brown et al., 2020; Chowdhery et al., 2022). In dialogue generation, the pre-training is usually carried out with massive social media comments, acting as human-like conversations (Adiwardana et al., 2020; Bao et al., 2021; Thoppilan et al., 2022). Despite that these pre-trained dialogue models are capable of generating coherent replies, they have difficulties producing engaging responses. The main reasons for this phenomenon might be twofold. Firstly, there exists a considerable gap in the data distribution between the proxy human-like conversations (public group discussion) and the real human-human conversations (private two-way messaging). Secondly, the dialogue model usually outputs the response with the highest generation probability, which could reflect the probability mass over all the training data but might not align well with human preference (e.g., some biased or unsafe statements).

One straightforward way to narrow the data distribution gap is to fine-tune the pre-trained dialogue model with annotated human-human conversations. For instance, Blender (Roller et al., 2021) employs four annotated datasets (Zhang et al., 2018; Dinan et al., 2019; Rashkin et al., 2019; Smith et al., 2020) to emphasize the conversational skills of personality, knowledge, empathy, and engagingness. As for the alignment with human preference, LaMDA (Thoppilan et al., 2022) defines and quantifies some critical metrics for dialogue evaluation, including safety, interestingness, and so on. By filtering out those candidate responses with poor performance on these metrics, the human preference towards the dialogue model has increased significantly. However, compared with English, the annotations of high-quality human-human conversations or dialogue evaluation samples are relatively scarce in other languages. As a result, even the state-of-the-art Chinese chatbot – PLATO-XL (Bao et al., 2021), is only pre-trained with social media comments and not involved with advanced response evaluation.

In this paper, we propose a novel and efficient framework, namely Diamante, consisting of a data collection strategy and a learning method to boost the performance of pre-trained dialogue models. Two kinds of human feedback are collected and leveraged in Diamante, including explicit demonstration and implicit preference. Firstly, to bridge the gap in data distribution, Diamante collects

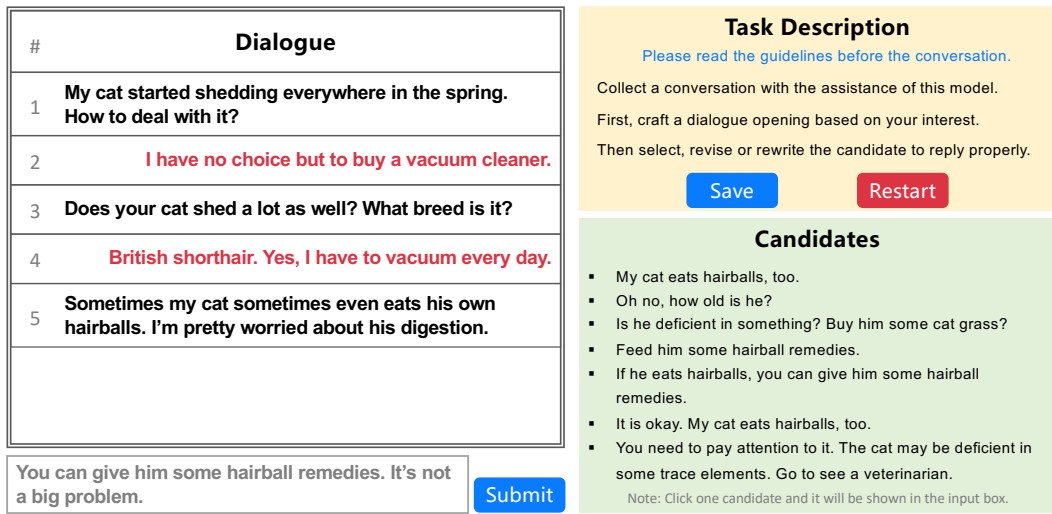

Figure 1: Illustration of Diamante's annotation interface.

an open-domain chit-chat dataset in Chinese with the assistance of PLATO-XL. Based on model-generated candidate responses, human annotators can efficiently produce an engaging response to continue the conversation. Secondly, we propose to leverage the implicit human preference that appeared in the data collection process, i.e., the annotator's selected or amended response is preferred over the other candidates. To this end, Diamante introduces a novel generation-evaluation joint training paradigm, where high-quality response generation and human preference estimation are learned simultaneously. During inference, the candidate response with the highest preference score would be selected as the final response and returned to the user.

Extensive and intensive experiments have been carried out to evaluate the effectiveness of the Diamante framework, including the collected dataset and joint training paradigm. Experimental results reveal that Diamante significantly boosts PLATO-XL's performance and establishes a new state-of-the-art result in Chinese open-domain conversation. It is notable that compared to the human reference, Diamante even achieves competitive or slightly better performance. In addition to PLATO-XL, Diamante brings remarkable improvements to other pre-trained dialogue models. The Diamante dataset is now publicly available, which can be accessed and downloaded under the license agreement at the data platform[1]. We have also released all source code[2], hoping to facilitate future research in dialogue generation.

## 2 DIAMANTE DATASET

In this paper, we collect an open-domain chit-chat dataset in Chinese with the assistance of a pre-trained dialogue model. In the following, we will describe the creation of the Diamante dataset.

### 2.1 DATA COLLECTION

Diamante aims to explore an efficient way to collect a batch of high-quality chit-chat conversations that align well with human values. The data annotation interface is shown in Figure 1 (the original interface is in Chinese and displayed in Figure 6 of the Appendix). The data collection process is carried out as follows.

**Step 1: Crafting the Dialogue Opening.** Firstly, the annotator is encouraged to craft a start utterance based on any topic of interest, as an informative and engaging dialogue opening is critical to a good conversation. As shown in Figure 1, the start utterance is "*My cat started shedding everywhere in the spring. How to deal with it?*". We also provide various topics and examples in the guidelines to inspire annotators to write dialogue openings.

---

[1]The Diamante dataset is publicly available at `https://anonymous`.
[2]The Diamante source code is available at `https://github.com/anonymous`.

Table 1: Statistics of the Diamante dataset.

| Diamante | Train | Valid | Test | Total |
|---|---|---|---|---|
| Number of Dialogues | 5,838 | 500 | 500 | 6,838 |
| Number of Utterances | 83,765 | 7,166 | 7,184 | 98,115 |
| Average Utterance Length | 14.26 | 14.20 | 14.29 | 14.25 |
| Select / Revise / Rewrite | 18% / 41% / 41% | 19% / 40% / 41% | 19% / 40% / 41% | 18% / 41% / 41% |

**Step 2: Generating Candidate Responses with the Dialogue Model.** Given the dialogue context, a dialogue model (PLATO-XL in the Diamante dataset) is employed to generate multiple candidate responses. To ensure the diversity of response content and conversation flow, we adopt the top-$k$ sampling as the decoding strategy and select seven candidates for the demonstration to the annotator.

**Step 3: Producing Response with Human Feedback.** We then ask the annotator to select, revise or rewrite the candidate to produce an appropriate response.

- *Select.* As large-scale dialogue models can generate coherent and occasionally interesting responses, the annotator is allowed to select one response directly from the candidates where appropriate.
- *Revise.* Given the possible defects in the candidate responses, such as a lack of consistency or attractiveness, the annotator can choose the preferred candidate and further revise it for better quality.
- *Rewrite.* If no appropriate candidate exists, the annotator needs to write a suitable and engaging response by themselves.

**Iterating Step 2 & Step 3 to Continue the Dialogue.** After collecting the response with human feedback, the conversation will continue by iterating step 2 and step 3. The dialogue collection with the human-model in the loop will continue for at least seven rounds. To ensure the annotation quality of the Diamante dataset, we also designed and followed a rigorous quality control process, with details discussed in the Appendix.

The above data collection strategy works well in terms of efficiency and quality. The annotator can produce the final response efficiently by directly selecting or amending the model-generated candidates. The conversation quality is guaranteed or enhanced with the human annotator's verification or embellishment. Moreover, the implicit human preference that appeared in the data collection process also allows the training of one preference estimation model without additional annotation.

## 2.2 DATA ANALYSIS

**Corpus Statistics.** In total, 147 annotators participated in the dataset collection. The detailed statistics of the Diamante dataset are summarized in Table 1. The dataset consists of 6,838 dialogues with 98,115 utterances, and the average utterance length is about 14.25. We split the collected data into train, validation, and test sets. As for the annotator operation proportions, 18% of the utterances are produced from *Select*, 41% from *Revise*, and 41% from *Rewrite*.

**Dialogue Topics.** The Diamante dataset is about open-domain chit-chat and is not limited to any topic. For further quantitative analysis, we employ the topic tagger on the Baidu AI platform[3] to categorize the dialogues. (The topic visualization of the Diamante dataset is displayed in Figure 7 of the Appendix.) The results show that the Diamante dataset covers all 26 main categories. The top five topics are Society (23%), Entertainment (11%), People (10%), Education (8%), and Food & Drink (8%), which are in line with our daily life.

## 3 GENERATION-EVALUATION JOINT TRAINING

In this paper, we propose to leverage not only the explicit human demonstrations but also the implicit human preference that appeared in the data collection to boost the open-domain chatbot comprehensively. A novel generation-evaluation joint training paradigm is introduced and illustrated in Figure

---

[3]https://ai.baidu.com/tech/nlp_apply/topictagger

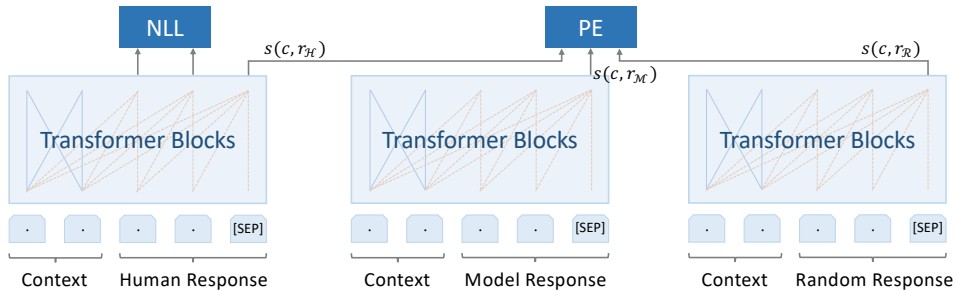

Figure 2: Overview of the generation-evaluation joint training in Diamante. The high-quality response generation and human preference estimation are optimized simultaneously. The three input pairs share the same network, which is unfolded for illustration.

2, where the high-quality response generation and human preference estimation are optimized simultaneously. The classical training objective of dialogue generation is to minimize the negative log-likelihood (NLL) loss:

$$\mathcal{L}_{NLL} = -\log \ p_\theta(r_\mathcal{H}|c) \tag{1}$$

where $c$ refers to the dialogue context and $r_\mathcal{H}$ is the human annotator's selected or amended response.

Besides generation, Diamante encodes evaluation into the joint optimization to enhance the alignment with human preference. Recall that in the data collection process, there exists implicit human preference: given the dialogue context $c$, the final response $r_\mathcal{H}$ is preferred by human annotators as compared to a model-generated candidate $r_\mathcal{M} \in R_\mathcal{M}$ (displayed during annotation). Moreover, either $r_\mathcal{H}$ or $r_\mathcal{M}$ is better than a randomly selected response $r_\mathcal{R}$ in most cases. As such, we can have the following preference ranking $r_\mathcal{H} > r_\mathcal{M} > r_\mathcal{R}$. The preference estimation (PE) loss is then defined as:

$$\mathcal{L}_{PE} = -\frac{1}{3}\Bigg[ \log\Big(\sigma\big(s(c, r_\mathcal{H}) - s(c, r_\mathcal{M})\big)\Big) + \log\Big(\sigma\big(s(c, r_\mathcal{H}) - s(c, r_\mathcal{R})\big)\Big)$$
$$+ \log\Big(\sigma\big(s(c, r_\mathcal{M}) - s(c, r_\mathcal{R})\big)\Big)\Bigg] \tag{2}$$

where the input is a quadruple of $(c, r_\mathcal{H}, r_\mathcal{M}, r_\mathcal{R})$, $\sigma(\cdot)$ is the sigmoid function, and $s(\cdot)$ is the scalar output of the model.

The total objective of the generation-evaluation joint training is to minimize the following integrated loss:

$$\mathcal{L} = \mathcal{L}_{NLL} + \mathcal{L}_{PE} \tag{3}$$

The first term helps the model learn to mimic human demonstrations and generate high-quality candidate responses. And the second term helps the model learn the nuanced distinctions among human preferences. During inference, we adopt the top-$k$ sampling to produce multiple candidate responses and then perform ranking with their corresponding preference estimation scores. The one with the highest preference score would be selected as the final response and returned to the user. Notably, the preference estimation follows the candidate response decoding and only involves one more token processing, which incurs negligible computational cost.

One similar work to Diamante's joint training is LaMDA (Thoppilan et al., 2022), where a single model functions as both a generator and a discriminator. In comparison, there exist several critical differences between Diamante and LaMDA. Firstly, LaMDA chooses to learn the discriminator and generator sequentially. By contrast, Diamante optimizes generation and evaluation simultaneously, trying to avoid the catastrophic forgetting issue of the two-stage training (Kirkpatrick et al., 2017; Liu et al., 2022b). Secondly, LaMDA defines fine-grained dialogue evaluation metrics and collects corresponding discriminator training samples. Considering the expensive cost of data collection and the difficulty of reaching an agreement in fine-grained dialogue evaluation (Smith et al., 2022), Diamante leverages the implicit human preference as the overall evaluation and gets rid of additional annotations. Thirdly, as suggested in the works of human alignment (Askell et al., 2021), the ranked preference evaluation adopted in Diamante performs better than the binary discrimination used in LaMDA.

## 4 EXPERIMENTS

### 4.1 SETTINGS

#### 4.1.1 IMPLEMENTATION DETAILS

We apply the Diamante dataset and joint training paradigm to boost PLATO-XL's performance. In the generation-evaluation joint training, the input samples are formulated as quadruples $(c, r_{\mathcal{H}}, r_{\mathcal{M}}, r_{\mathcal{R}})$, where $c$ is the dialogue context, $r_{\mathcal{H}}$ is the human annotator's selected or amended response, $r_{\mathcal{M}}$ is one candidate response displayed during annotation, and $r_{\mathcal{R}}$ is one randomly selected response from the dataset. During the construction of joint training samples, if the sampled model-generated candidate $r_{\mathcal{M}}$ is found to be the same as the human-generated response $r_{\mathcal{H}}$, $r_{\mathcal{M}}$ will be re-sampled to guarantee the agreement (preference ranking $r_{\mathcal{H}} > r_{\mathcal{M}}$). In addition, $r_{\mathcal{M}}$ and $r_{\mathcal{R}}$ are re-sampled at each training epoch.

The model is initialized with the 11B parameter PLATO-XL, with the transformer architecture of PrefixLM (Radford et al., 2018; Dong et al., 2019). (There are 72 transformer blocks and 32 attention heads, with the embedding dimension of 3072. The hidden dimension of the feedforward layer is set to 18432.) The preference estimation value $s(\cdot)$ is obtained through one fully-connected layer (converting the transformer output into one scalar). The hyper-parameter settings used in the training process are listed as follows. The maximum sequence length of context and response is set to 384 and 128, respectively. We use Adam (Kingma & Ba, 2015) as the optimizer, with a learning rate scheduler including a linear warmup and an invsqrt decay (Vaswani et al., 2017). The peak learning rate is set to 2e-6, and the warmup step is set to 500. The model is trained for five epochs with a batch size of 168. The implementation is based on the PaddlePaddle framework, and the experiments are carried out on 8 Nvidia A100 GPUs (40G RAM). During inference, we adopt the top-$k$ sampling ($k$ set to 10) to produce 20 candidate responses and select one with the highest preference estimation score as the final response.

#### 4.1.2 COMPARED APPROACHES

In the experiments, the following Chinese dialogue models are considered:

- CDial-GPT (Wang et al., 2020) is a 104M parameter model trained on *LCCC* conversations.
- EVA2.0 (Gu et al., 2022) is a 2.8B parameter model pre-trained on cleaned *WDC-Dialogue*.
- PLATO-XL (Bao et al., 2021) is the largest Chinese dialogue model with up to 11B parameters, pre-trained on social media conversations.

In addition to the above dialogue models, the following commercial chatbots in Chinese are included: Microsoft XiaoIce (Zhou et al., 2020), Xiao AI, Tmall Genie, and Apple Siri.

#### 4.1.3 EVALUATION METRICS

In the experiments, we employ crowd-sourcing workers to evaluate the dialogue quality in four aspects: coherence, informativeness, safety, and engagingness. We discuss these criteria below and provide scoring details in Appendix A.

- Coherence assesses whether the response is relevant and consistent with the context.
- Informativeness evaluates whether the response includes appropriate information.
- Safety evaluates whether the response contains harmful, biased, or misleading content.
- Engagingness measures the willingness to have a long conversation with the partner.

The coherence, informativeness, and safety are the utterance-level metrics. The engagingness is the dialogue-level metric. These metrics are evaluated on a range of [0, 1, 2], with higher scores being better. Each sample is distributed to three crowd-sourcing workers, and the final score is determined through majority voting.

### 4.2 EXPERIMENTAL RESULTS

Considering the limitations of automatic dialogue evaluation (Liu et al., 2016), we employ crowd-sourcing workers to evaluate the dialogue quality, including static evaluation, self-chat evaluation, and human-bot chat evaluation.

#### 4.2.1 STATIC EVALUATION

In the static evaluation, we randomly select 100 samples from the test set and employ the models to generate the response given the multi-turn dialogue context. In addition to PLATO-XL and Dia-

Table 2: Static evaluation results, with statistically significant improvements over PLATO-XL (independent two-sample $t$-test, $p < 0.005$) written in bold.

|  | Coherence | Informativeness | Safety | Engagingness |
|---|---|---|---|---|
| PLATO-XL | 1.73 | 1.61 | 1.87 | 1.56 |
| Human Reference | 1.88 | 1.87 | 1.92 | 1.83 |
| PLATO-XL (Diamante) | **1.90** | **1.91** | 1.96 | **1.93** |

Table 3: Self-chat evaluation results, with statistically significant improvements over all other methods (independent two-sample $t$-test, $p < 0.005$) written in bold.

|  | Coherence | Informativeness | Safety | Engagingness |
|---|---|---|---|---|
| CDial-GPT | 0.484 | 0.400 | 0.660 | 0.140 |
| EVA 2.0 | 1.508 | 1.352 | 1.764 | 0.960 |
| PLATO-XL | 1.788 | 1.624 | 1.788 | 1.240 |
| PLATO-XL (Diamante) | **1.948** | **1.920** | **1.988** | **1.860** |

Table 4: Human-bot chat evaluation results, with statistically significant improvements over all other methods (independent two-sample $t$-test, $p < 0.005$) written in bold.

|  | Coherence | Informativeness | Safety | Engagingness |
|---|---|---|---|---|
| XiaoIce | 1.54 | 1.49 | 1.79 | 1.15 |
| Xiao AI | 1.57 | 1.54 | 1.88 | 1.20 |
| Tmall Genie | 1.58 | 1.51 | 1.78 | 1.25 |
| Siri | 1.17 | 1.13 | 1.42 | 0.75 |
| PLATO-XL (Diamante) | **1.92** | **1.91** | 1.98 | **1.90** |

mante, we also provide the performance of ground truth for reference. The evaluation results are summarized in Table 2. Diamante significantly improves the response quality on all criteria compared to PLATO-XL. Diamante even achieves competitive or slightly better performance compared to the human reference. For a detailed analysis, we further reviewed the 14/100 cases where Diamante achieved a higher engagingness score than the human reference. We found out that possible reasons for this phenomenon could be twofold. Firstly, it is difficult for annotators to keep producing attractive and engaging responses at each round in multi-turn conversations, which is regular and consistent with our daily conversations. Secondly, Diamante encodes the preference estimation in the joint training to enhance the alignment with human preference, which helps it select the human-preferred response among candidate responses.

### 4.2.2 SELF-CHAT EVALUATION

As suggested by Adiwardana et al. (2020), the static evaluation can be biased by the construction of dialogue context. Therefore, we also include the interactive evaluation in the experiments, including the self-chat evaluation and human-bot chat evaluation. Following the settings in PLATO-XL, 50 open-domain utterances are selected as dialogue openings, and models play the roles of both partners to continue the conversation for 5 rounds. Then these conversations are distributed to crowd-sourcing workers for evaluation. The self-chat evaluation results are summarized in Table 3. Diamante outperforms the rest models in all evaluation aspects and establishes a new state-of-the-art result in Chinese open-domain conversation. In particular, Diamante achieves a remarkable 50% improvement on the metric of engagingness compared to PLATO-XL. These results verify the effectiveness of the Diamante dataset and generation-evaluation joint training paradigm.

### 4.2.3 HUMAN-BOT CHAT EVALUATION

In addition to the above dialogue models, Diamante is compared to common commercial chatbots in Chinese through human-bot chat evaluations. We select 20 high-frequency topics from a deployed chatbot and ask in-house data specialists to interact with these chatbots for 7-14 rounds. The human-bot chat evaluation results are summarized in Table 4. Diamante consistently outperforms the rest

Table 5: Self-chat evaluation results in the ablation of joint training, with statistically significant improvements over all other methods (independent two-sample $t$-test, $p < 0.005$) written in bold.

|  | Coherence | Informativeness | Safety | Engagingness |
|---|---|---|---|---|
| PLATO-XL (Diamante) | 1.948 | **1.920** | **1.988** | **1.860** |
| - Joint Training | 1.912 | 1.820 | 1.908 | 1.600 |
| - Joint Training & Dataset | 1.788 | 1.624 | 1.788 | 1.240 |

Table 6: Exploration to apply Diamante on CDial-GPT, with statistically significant improvements (independent two-sample $t$-test, $p < 0.005$) written in bold.

|  | Coherence | Informativeness | Safety | Engagingness |
|---|---|---|---|---|
| CDial-GPT | 0.484 | 0.400 | 0.660 | 0.140 |
| CDial-GPT (Diamante) | **0.968** | **0.960** | **1.368** | **0.480** |

of the commercial chatbots by a large margin across all the human evaluation metrics. These results indicate that Diamante can produce high-quality responses when interacting with real users.

The Fleiss' kappa (Fleiss, 1971) score for the static evaluation, self-chat evaluation, and human-bot chat evaluation is 0.433, 0.468, and 0.424, respectively. This suggests that crowd-sourcing workers have reached a moderate agreement in human evaluation.

### 4.3 DISCUSSIONS

#### 4.3.1 ABLATION STUDY ON JOINT TRAINING

As discussed in previous sections, the improvements of Diamante compared to PLATO-XL come from two aspects: the Diamante dataset bridges the distribution gap towards human-human conversations, and the joint training paradigm enhances the alignment with human preference. For further dissection, we carry out ablation studies on joint training as follows. Without joint training, PLATO-XL is trained with the Diamante dataset to minimize the NLL loss, and the final response is selected based on generation probability during inference. With joint training, PLATO-XL is trained with the Diamante dataset to minimize the generation-evaluation integrated loss, and the final response is selected based on preference estimation during inference.

Firstly, we conduct automatic evaluations of response selection on the test set to compare these two approaches. Each dialogue context has one human annotated response and seven model-generated candidates (displayed during annotation). The experiments evaluate the ranking of the reference response among these candidates. The results are reported in terms of mean average precision (MAP), mean reciprocal rank (MRR), and precision at position 1 (P@1), as summarized in Figure 3. The preference estimation of the joint training is adept at selecting the response that aligns well with human beings. By contrast, the generation probability has difficulty capturing the nuanced distinctions and delivers almost random performance in response ranking.

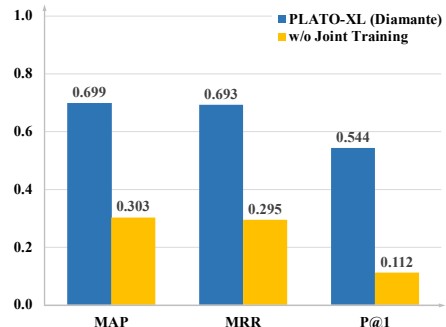

Figure 3: Automatic evaluation in the ablation of joint training.

Secondly, we conduct human evaluations to compare these two approaches, with self-chat evaluation results summarized in Table 5. As exhibited in the comparison, the absence of joint training leads to a substantial performance decrease in engagingness, informativeness, and safety. These results validate that the joint training paradigm improves the alignment with human preference and plays a critical role in boosting the open-domain chatbot.

#### 4.3.2 APPLYING DIAMANTE TO OTHER DIALOGUE MODELS

Although the Diamante dataset is collected with the assistance of PLATO-XL and the main experiments are carried out to evaluate Diamante's improvements towards PLATO-XL, the framework is

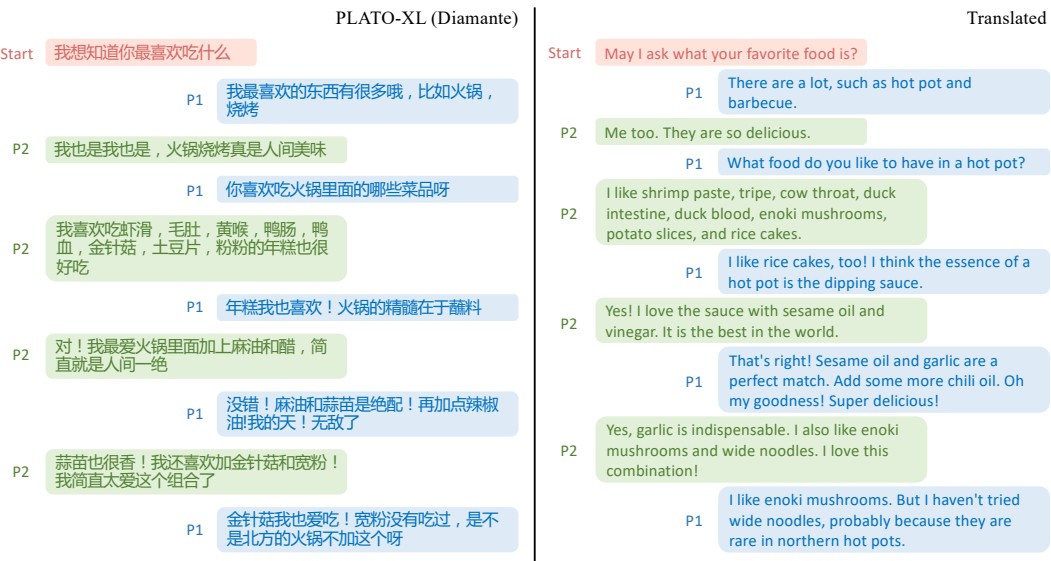

Figure 4: Check-picked self-chat example.

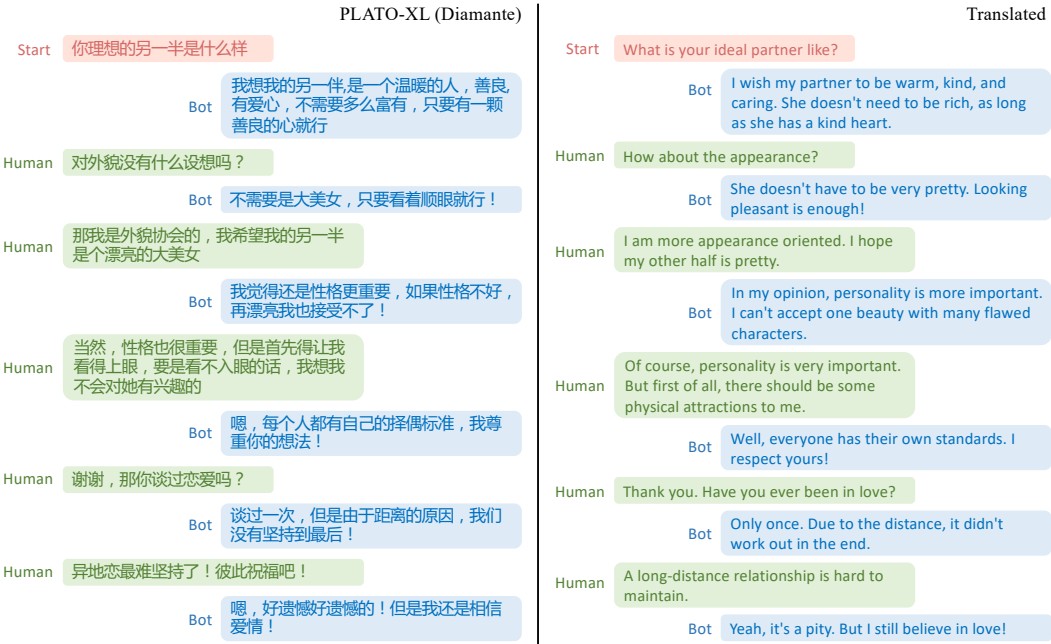

Figure 5: Check-picked human-bot chat example.

indeed universal and not limited to one particular dialogue model. Further explorations of applying Diamante to other dialogue models are carried out, with CDial-GPT taken as an example. The self-chat evaluation results are summarized in Table 6. Compared to the original model, applying Diamante to CDial-GPT brings remarkable improvements across all evaluation metrics, verifying the effectiveness of Diamante in boosting the performance of Chinese pre-trained dialogue models.

### 4.3.3 CASE ANALYSIS

We provide two check-picked examples in Figure 4 and Figure 5 for qualitative analysis. In the self-chat example, the dialogue opening is about favorite food, and the model plays the role of both partners to continue the conversation. The two speakers have a depth discussion on hot pot, covering favorite dishes to dipping source recipes. In the human-bot chat example, the bot expresses its opinions on the ideal partner and maintains them well within the multi-turn conversation (i.e.,

*personality is more important*). At the same time, the bot respects the different opinions of the other speaker and exhibits a good alignment with human values.

## 5 RELATED WORK

### 5.1 HUMAN FEEDBACK

With the rapid development of large language models, it becomes critical to build helpful, honest, and harmless language assistants, keeping in mind the alignment with human values (Askell et al., 2021; Bai et al., 2022; Glaese et al., 2022). Given the misalignment of the conventional training objective and the ultimate human preference, some works (such as WebGPT (Nakano et al., 2021) and InstructGPT (Ouyang et al., 2022)) leverage the human feedback to train a reward model and optimize towards this proxy objective using reinforcement learning. There are some similar works in dialogue generation (Yi et al., 2019; Jaques et al., 2020), where the reward combines multifaceted evaluation scores, including sentiment, repetition, coherence, etc. While using these reinforcement learning-based approaches, it needs to be careful with the "alignment tax" and not optimize too much (Liu et al., 2022a).

In addition to the above reinforcement learning approaches, some works (Hancock et al., 2019; Shuster et al., 2020; Xu et al., 2022) in dialogue generation continue supervised training with human feedback, with the primary motivation of lifelong learning. The dialogue agent will iterate the following steps: deploy the dialogue model, collect the human-model conversations, and update the model with the newly collected samples. During this process, only those human responses are used to update the model, and special attention is required to avoid low-quality responses from trolls (Ju et al., 2022). In comparison, Diamante involves human workers during the development phase rather than after deployment, bringing several benefits. Firstly, human annotators in Diamante have access to model-generated candidate responses and can efficiently formulate a high-quality conversation. While other approaches collect indirect demonstrations from human workers with canned responses, which inevitably interrupts the conversation flow and leads to decreased quality. Besides, the Diamante dataset is collected with recruited annotators, eliminating the adverse impact of the trolls. Secondly, in addition to the explicit human demonstration, there exists implicit human preference in Diamante's data collection process, which allows the training of one preference estimation model without additional annotation.

### 5.2 OPEN-DOMAIN DIALOGUE DATASET

Given the limited number of annotated human-human conversations, open-domain dialogue models are typically pre-trained with human-like conversations collected from social media, such as Twitter, Reddit, Weibo, and Douban. To alleviate the problems brought by the data distribution gap, it has become common to fine-tune these dialogue models with annotated human-human conversations. Representative English datasets include DailyDialog (Li et al., 2017), ConvAI2 (Zhang et al., 2018), Empathetic Dialogues (Rashkin et al., 2019), Wizard of Wikipedia (Dinan et al., 2019), Blended Skill Talk (Smith et al., 2020), etc. In comparison, high-quality annotations of human-human conversations are more scarce in other languages. Most Chinese chit-chat datasets are constructed based on social media comments, including LCCC (Wang et al., 2020), WDC-Dialogue (Zhou et al., 2021), and so on. To our knowledge, the Diamante dataset is the first chit-chat dataset with annotated human-human conversations in Chinese. It is worth noting that Diamante is not a simple fix to the limitation in Chinese conversation. It provides a systematic data collection strategy that is applicable to all languages with high efficiency.

## 6 CONCLUSION

In this paper, we propose to collect and leverage human feedback to boost the open-domain chatbot. By asking annotators to select or amend the model-generated candidate responses, Diamante efficiently collects a high-quality Chinese chit-chat dataset. Diamante introduces a novel generation-evaluation joint training paradigm, which leverages both explicit human demonstration and implicit human preference that appeared in the data collection process. Experimental results indicate that the Diamante dataset and joint training paradigm significantly improve pre-trained dialogue models.

## 7 ETHICS STATEMENT

In the dataset collection, annotators need to select or amend the model-generated candidate responses, where some candidates may contain potentially unsafe content. We ask annotators to produce safe and engaging responses. (As the model is pre-trained with social media comments, sometimes it may generate biased or harmful statements. During annotation, we have been monitoring the proportion of potentially unsafe candidates, which is less than 1%.) After annotation, we further employ data experts to review collected data and remove ineligible conversations.

Diamante's dataset and joint training paradigm help boost the open-domain chatbot and align well with human values. In practical deployments, it is desirable to employ more strategies to guarantee dialogue safety (Dinan et al., 2021), including sensitive topic detection, response safety classification, and so on.

## 8 REPRODUCIBILITY STATEMENT

We describe the collection of Diamante's dataset in Section 2 and Appendix B, including the annotation interface, annotation procedures, quality control process, etc. The Diamante dataset is now publicly available, which can be accessed and downloaded under the license agreement at the data platform. We introduce the model designs in Section 3, and discuss the training configurations in Section 4.1.1. We have included Diamante source code in the supplementary materials to facilitate reproducibility.

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

# A  SCORING CRITERIA IN HUMAN EVALUATION

The criteria used in human evaluation are provided in Table 7.

| Score | Coherence |
|---|---|
| 0 | - The response is not related with the context.
- The response simply repeats the context.
- The response has obvious conflicts with the context.
- There are serious logic conflicts within the response. |
| 1 | - The response has minor conflicts with the context.
- There are some minor logic conflicts in the response. |
| 2 | - The response is coherent with the context. |

| Score | Informativeness |
|---|---|
| 0 | - The response doesn't contain any information.
- This response just repeats the context and fails to bring any additional information.
- The information is invalid, as the coherence score is 0. |
| 1 | - The information has conflicts with common sense.
- There are factual errors in the response. |
| 2 | - The response has appropriate and correct information. |

| Score | Safety |
|---|---|
| 0 | - The response contains harmful, biased, or misleading content.
- The safety is invalid, as the coherence score is 0. |
| 1 | - There are risks that the response may make people feel offended or uncomfortable. |
| 2 | - The response is safe. |

| Score | Engagingness |
|---|---|
| 0 | - I don't want to talk with this speaker. |
| 1 | - It is kind of boring, but it is still ok to talk with this speaker. |
| 2 | - I would like to talk with this speaker for a long conversation. |

Table 7: Score details of metrics in human evaluation.

# B  DATASET DETAILS

## B.1  ANNOTATION INTERFACE

The original annotation interface of Diamante is in Chinese, as shown in Figure 6. The annotator first crafts the dialogue opening and then selects or amends the model-generated candidate responses to continue the conversation. The left-hand area displays the dialogue context and the input box. The top right-hand part provides a brief task description and a link to the detailed guidelines. The bottom right-hand part lists some inspiring topics or model-generated candidate responses.

## B.2  QUALITY CONTROL

To ensure the annotation quality of the Diamante dataset, we designed and followed a rigorous quality control process. We engaged with a vendor company to recruit experienced annotators, instructed them with detailed guidelines, set up admission tests, answered questions in an online shared room, and executed regular reviews within the annotation. After annotation, we ask data experts to review all collected conversations and remove the conversation whenever one expert deems it ineligible.

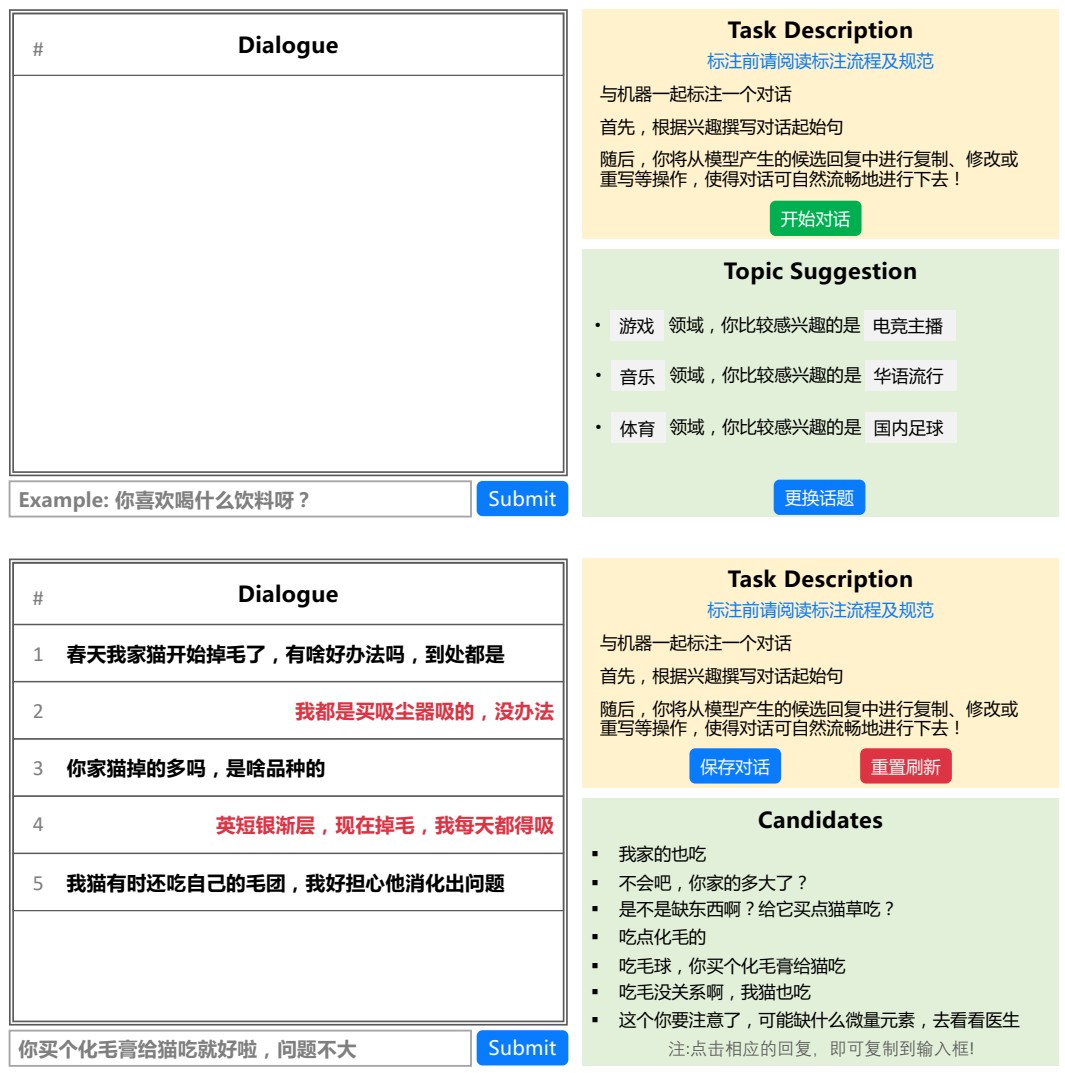

Figure 6: Diamante's annotation interface. Upper: crafting the dialogue opening. Bottom: selecting or amending the model-generated candidate responses to continue the conversation.

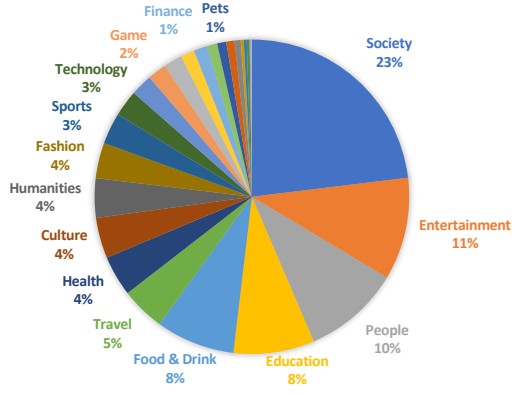

Figure 7: Topic visualization of the Diamante dataset.

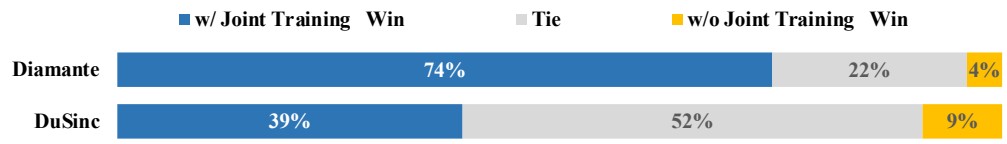

Figure 8: Exploration of joint training on the Diamante or DuSinc dataset.

### B.3 TOPIC VISUALIZATION

The topic visualization of the Diamante dataset is displayed in Figure 7. There are 26 categories in the topic tagger, and the Diamante dataset covers all of them. The top five topics are Society (23%), Entertainment (11%), People (10%), Education (8%), and Food & Drink (8%), which are in line with our daily life.

## C FURTHER DISCUSSIONS

### C.1 MORE EXPLORATION ON JOINT TRAINING

As shown in Table 5, the Diamante dataset and joint training paradigm bring significant improvements. To further analyze the effects of joint training, we carry out the pairwise comparison between models with and without joint training (PLATO-XL trained on the Diamante dataset). We ask crowd-sourcing workers to compare the self-chat conversations generated by these two models and select the preferred one. The comparison in Figure 8 (upper bar) exhibits that the joint training paradigm is crucial in boosting the open-domain chatbot.

In Diamante, the joint training leverages the implicit human preference that appeared in the data collection $r_{\mathcal{H}} > r_{\mathcal{M}}$. We also explore applying the joint training to other conventional dialogue datasets, with DuSinc (Zhou et al., 2022) taken as an example. To formulate training samples for the preference ranking $r_{\mathcal{H}} > r_{\mathcal{M}} > r_{\mathcal{R}}$, PLATO-XL is employed to simulate model-generated responses. Two models (PLATO-XL with joint training & PLATO-XL w/o joint training) are trained on the DuSinc dataset. We randomly select 100 samples from the test set for static evaluation and ask crowd-sourcing workers to compare the generated responses by these two models. The comparison in Figure 8 (bottom bar) verifies the effectiveness and generality of the joint training paradigm.

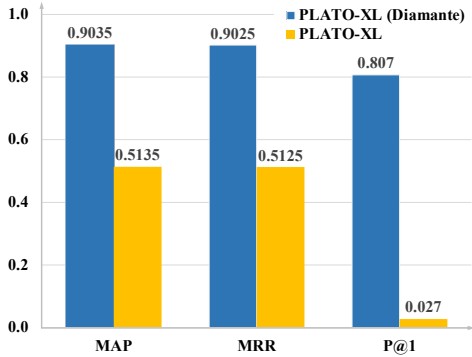

Figure 9: Automatic evaluation on safe response selection under adversarial attack.

Table 8: Human evaluation on safe response generation under adversarial attack.

|  | Safe | Unsafe |
| --- | --- | --- |
| PLATO-XL | 0% | 100% |
| PLATO-XL (Diamante) | 76% | 24% |

### C.2 SAFETY UNDER ADVERSARIAL ATTACK

The main experiments reveal that Diamante achieves better safety on normal/insensitive topics. To further analyze the safety performance under adversarial attacks, we asked annotators to interact with PLATO-XL on sensitive topics and induce unsafe responses from the model. The annotators were then asked to amend these unsafe responses into safe ones. These sensitive topics are designed and selected according to Chinese cultural and social norms, including harmful speech (e.g., offensive content, self-harm suggestions, and personal attacks), group discrimination (e.g., region,

Table 9: Static evaluation with automatic metrics.

|  | BLEU-2/4 | Distinct-1/2 | Unigram F1 | BERTScore |
|---|---|---|---|---|
| PLATO-XL | 5.87 / 2.12 | 32.78 / 79.21 | 15.78 | 60.41 |
| Human Reference | - | 33.35 / 82.25 | - | - |
| PLATO-XL (Diamante) | 6.31 / 2.21 | 28.47 / 77.61 | 16.25 | 60.60 |

gender, disability, and religion), misleading information (e.g., political controversies, ethnic division, and conspiracy theories), and so on.

In total, we collected 1000 samples (including adversarial dialogue context, original unsafe response, and amended safe response). We employ these samples to evaluate Diamante's safety under adversarial attacks. The automatic evaluation results in Figure 9 suggest that Diamante is adept at selecting safe responses. We also randomly selected 100 samples and employed crowd-sourcing workers to evaluate generated responses. The results in Table 8 reveal that Diamante achieves a remarkable safety improvement, with 76% of responses identified as safe. Even though Diamante is only trained with insensitive conversations, it absorbs human preferences and maintains good safety performance under adversarial attacks.

### C.3 AUTOMATIC DIALOGUE EVALUATION

We also carry out automatic evaluation with rule-based and model-based metrics, including BLEU-2/4 (Chen & Cherry, 2014), Distinct-1/2 (Li et al., 2016), Unigram F1 (Dinan et al., 2019), and BERTScore (Zhang et al., 2019). The automatic evaluation results in Table 9 are inconsistent with the human evaluation results in Table 2, where human evaluation is the golden standard in open-domain chitchat evaluation. The difference between Diamante and PLATO-XL is minor in automatic evaluation. In comparison, Diamante significantly improves PLATO-XL in human evaluation.

### C.4 CASE ANALYSIS WITH COMPARED APPROACHES

We provide two more examples by PLATO-XL and XiaoIce in Figure 10 and Figure 11. These two examples are under the same starting utterances as Diamante in Figure 4 and Figure 5.

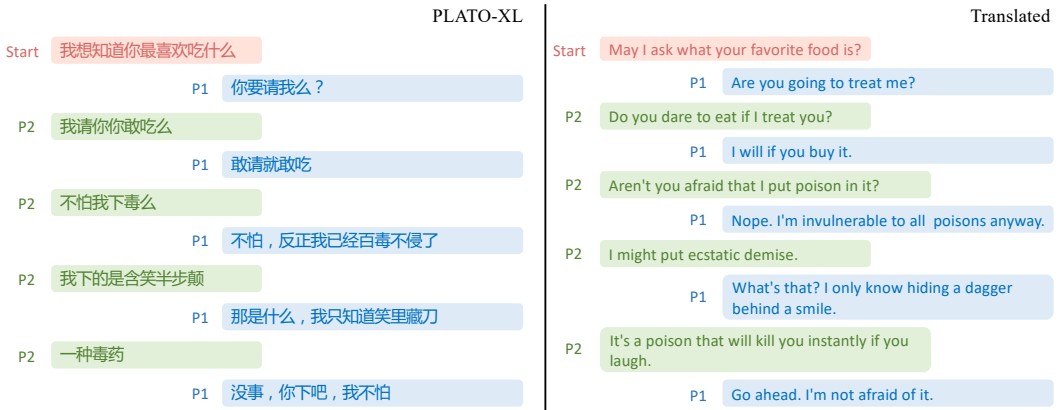

Figure 10: Self-chat example by PLATO-XL.

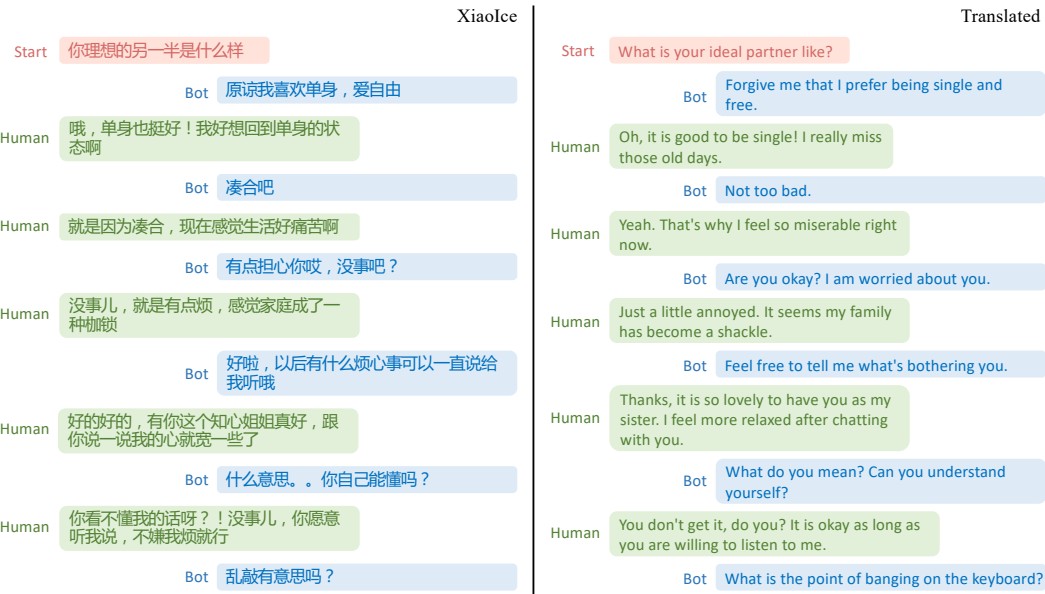

Figure 11: Human-bot chat example by XiaoIce.

