# OpenReview forum: "Towards Boosting the Open-Domain Chatbot with Human Feedback"
_ICLR.cc/2023/Conference — Submitted to ICLR 2023_

### Official Review · Reviewer_rtcq · 2022-10-26

**Confidence:** 4
**Correctness:** 3
**Technical Novelty And Significance:** 2
**Empirical Novelty And Significance:** 2
**Recommendation:** 3

**Clarity, Quality, Novelty And Reproducibility:**

Some important details are missing:
1. The architecture of your model. Even if you directly use the plato-XL as your backbone, a brief introduction to its architecture is necessary for completeness.
2. How do you implement the s() function in Eq.2?
3. You adopt top-K sampling to produce multiple candidates. So what is the value of K in your experiment? How many candidates are generated for each case?

The novelty of the proposed method is under question. The paper learns the human feedback information by jointly training a discriminator and a generator. Actually, generate-then-rerank is a popular paradigm and has been used in many prior works such as [1][2][3].

[1]CALIBRATING SEQUENCE LIKELIHOOD IMPROVES CONDITIONAL LANGUAGE GENERATION
[2]JOINT GENERATOR-RANKER LEARNING FOR NATURAL LANGUAGE GENERATION
[3]BRIO: Bringing Order to Abstractive Summarization


**Strength And Weaknesses:**

Strengths:
1. The paper is well-written and easy to follow.
2. The authors ablate several aspects of their method.
3. The method obtains good results in human evaluation.

Weakness:



1. A major proposal of the paper is to use human feedback to improve the performance of dialogue system. However, it is not a new idea in my opinion. There is rich literature[1-5] discussing how to continue learning from human feedback after deployment, which is closely related to your method.  Out of the domain of dialogue, there is also a large body of work studying the improvement of models from human feedback, such as summarization[7][6] or question answering[8]. They should at least be included in the related work, or the readers will have difficulty putting the work into its literature.

2. The evaluation protocol is not very convincing to me.
(1) Metrics: Only human evaluation is conducted and no automatic metrics are involved. I acknowledge that some automatic metrics like BLEU or ROUGE may be poorly correlated with human, but some neural metrics such as BERTscore should be considered. Furthermore, pure human evaluation renders the model performance hard to reproduce.
(2) Dataset: In the statistics evaluation, only 100 cases are sampled from the test set and evaluated. Why not use all the test set? Do you sample multiple times and conduct a repetitive experiment or just report the result from a single sampling? Why not perform and compare on some widely-used open-domain dialogue dataset such as Douban[9] or STC[10]?
(3) Baselines: PLATO-XL is up to 11B while EVA2.0 is 2.8B and CDial-GPT is 104M. So I doubt whether your comparison is fair. Why not implement your method on some smaller version of PLATO model?



[1]Learning from Dialogue after Deployment: Feed Yourself, Chatbot!
[2]Learning New Skills after Deployment: Improving open-domain internet-driven dialogue with human feedback
[3]Improving alignment of dialogue agents via targeted human judgements
[4]Training a Helpful and Harmless Assistant with Reinforcement Learning from Human Feedback
[5]BlenderBot 3: a deployed conversational agent that continually learns to responsibly engage
[6]Learning to summarize from human feedback
[7]Self-critiquing models for assisting human evaluators
[8]WebGPT: Browser-assisted question-answering with human feedback
[9]Sequential matching network: A new architecture for multi-turn response selection in retrieval-based chatbots
[10]Neural responding machine for short-text conversation.



**Summary Of The Paper:**

This paper proposes a new technique to enhance open-domain chatbot with human feedback information. The authors attribute the
less-engaging response to the distribution gap between human-human conversation and proxy human-machine conversation. Therefore, they design a data collection procedure to collect human revisions to machine response and construct a dataset based on that. Utilizing the established dataset, the authors adopt the classical generate-then-rerank paradigm to first generate multiple candidates and then select one.


**Summary Of The Review:**

The authors propose a data collection strategy to collect human feedback for an open-domain chatbot.  post-training an existing pre-trained dialogue model (PLATO-XL) with the collected data achieves some preliminary results. However, the proposed method is not original and the evaluation setup is defective. I would suggest that a major revision of the manuscript is needed to reach the publishing requirement of ICLR or other top-tier conferences.

---

> ### Author Response · Authors · 2022-11-19
> **Thank you! We have updated the paper to include the results of automatic evaluation and details of experiment settings.**
>
> We appreciate the feedback from the reviewer. We will give more clarifications in the following. We have revised the paper to include the results of automatic evaluation and details of experiment settings.
>
> > There is rich literature[1-5] discussing how to continue learning from human feedback after deployment, which is closely related to your method. Out of the domain of dialogue, there is also a large body of work studying the improvement of models from human feedback, such as summarization[7][6] or question answering[8]. They should at least be included in the related work, or the readers will have difficulty putting the work into its literature.
>
> We would like to point out that most of these works ([1][2][4][8]) have been discussed in our manuscript. In Section 5.1, we discuss related works on human feedback from two aspects: reinforcement learning with human feedback (dialogue and related tasks, [4][8] discussed within this part) and lifelong learning after deployment (dialogue systems, [1][2] discussed within this part). As for the rest, we didn't include [5] because its relevance is already covered by [2]. Thanks for the suggestion. In this revision, we have included this recent work [3].
>
> > Metrics: Only human evaluation is conducted and no automatic metrics are involved. I acknowledge that some automatic metrics like BLEU or ROUGE may be poorly correlated with human, but some neural metrics such as BERTscore should be considered. Furthermore, pure human evaluation renders the model performance hard to reproduce.
>
> As suggested by the reviewer, we include the automatic evaluation with rule-based and model-based metrics in this revision, including BLEU-2/4, Distinct-1/2, Unigram F1, and BERTScore. The automatic evaluation results (Table 9 of the Appendix) are inconsistent with the human evaluation results (Table 2). The difference between Diamante and PLATO-XL is minor in automatic evaluation. In comparison, Diamante significantly improves PLATO-XL in human evaluation (independent two-sample $t$-test, $p < 0.005$).
>
> Notably, human evaluation is the golden standard in open-domain chitchat evaluation. Nowadays, large-scale open-domain chatbots mostly rely on human evaluation, including LaMDA, BlenderBot 3, PLATO-XL, etc. There might be several reasons for the rare adoption of automatic evaluation in open-domain chitchat.
>
> Firstly, the correlations of rule-based metrics (like BLEU, Distinct, and Unigram F1) and human evaluation become worse given the one-to-many mapping relationship in open-domain chitchat  (one dialogue context can have multiple appropriate responses). Secondly, the model-based metrics have difficulty assessing much more powerful dialogue models (e.g., Diamante is about 100 times larger than the BERTScore model). Thirdly, most automatic metrics rely on golden responses as references, which are only available in static evaluation. Given static evaluation can be biased by the construction of dialogue context, interactive evaluation is the mainstream in open-domain chitchat.
>
> We acknowledge the importance of automatic evaluation in reproducing the model performance. We hope our community works together towards automatic evaluation and makes some breakthroughs in open-domain conversation soon.
>
> > Why not perform and compare on some widely-used open-domain dialogue dataset such as Douban[9] or STC[10]?
>
> Thanks for raising this point. Since the datasets of Douban and STC are collected directly from public social media, most of these conversations are contained in the PLATO-XL's pre-training corpus (1.2B conversations from Chinese public social media). Considering the issue of data contamination, it is not appropriate to use these datasets for evaluation.
>
> To avoid potential data contamination, we use the test set of the newly collected Diamante dataset for static evaluation. Moreover, in our experiments, we employ less affected interactive evaluation in addition to static evaluation.

---

> > ### Author Response · Authors · 2022-11-19
> > **Continuation of the response**
> >
> > > PLATO-XL is up to 11B while EVA2.0 is 2.8B and CDial-GPT is 104M. So I doubt whether your comparison is fair. Why not implement your method on some smaller version of PLATO model?
> >
> > In this paper, Diamante aims to boost the performance of Chinese chatbots and chooses the state-of-the-art model at that time (PLATO-XL) for dataset collection. The main experiments are about the comparison between PLATO-XL and PLATO-XL (Diamante). The only difference between them is the Diamante dataset and joint training paradigm. They have the same scale, and the comparison between them is fair.
> >
> > In our manuscript (Section 4.3.2), we also explored applying Diamante to other pre-trained dialogue models, with CDial-GPT taken as an example. The evaluation results indicate that Diamante brings remarkable improvements to CDial-GPT across all evaluation metrics, verifying the effectiveness of Diamante in boosting the performance of Chinese pre-trained dialogue models.
> >
> > > Some important details are missing: 1. The architecture of your model. Even if you directly use the plato-XL as your backbone, a brief introduction to its architecture is necessary for completeness. 2. How do you implement the s() function in Eq.2?  3. You adopt top-K sampling to produce multiple candidates. So what is the value of K in your experiment? How many candidates are generated for each case?
> >
> > Thanks for catching these. In this revision, we have included these details.
> > - The baseline PLATO-XL adopts the transformer architecture of PrefixLM. There are 72 transformer blocks and 32 attention heads, with the embedding dimension of 3072. The hidden dimension of the feedforward layer is set to 18432.
> > - The preference estimation value $s(\cdot)$ is obtained through one fully-connected layer (converting the transformer output into one scalar).
> > - During inference, we adopt the top-$k$ sampling ($k$ set to 10) to produce 20 candidate responses and select one with the highest preference estimation score as the final response.

---

### Official Review · Reviewer_ftAn · 2022-10-27

**Confidence:** 4
**Correctness:** 3
**Technical Novelty And Significance:** 2
**Empirical Novelty And Significance:** 2
**Recommendation:** 5

**Clarity, Quality, Novelty And Reproducibility:**

The paper is very well written and reproducible (code will be released). There's very interesting NLP contribution here but less novelty in terms of learning representation methods used. The work takes inspiration from the LaMDA model training strategies but claims to overcome the identified gaps in the LaMDA model. The paper uses standard generation strategies (top-K sampling instead of others such as diversity based sampling methods in literature to encourage more diverse model responses).



**Strength And Weaknesses:**

- the paper uses standard ML methods for the training strategies without any significant technical innovation of the work.
- However, the model training setup is pretty interesting, combining mask learning with contrastive learning using implicit human preferences.
- the model training does not explicitly seem to leverage the revisions which would have been interesting to model.
- the step 1 (crafting the dialog opening) isn't very innovative. Authors mention they suggest topics in the guidelines but there isn't any AI assisted or explicit strategy to encourage users to discuss diverse topics.
- addition AI assistance in the interface could help with faster data collection (sentence completion in the text box for example and if users select or reject the sentence level word completions)
- all the results suggest that this model is better on all the studies metrics in all the comparisons. This is a very strong result; however similar analysis on another dataset (maybe another language) would convince the readers much better on the strength of the model fine-tuning.



**Summary Of The Paper:**

The paper proposes a a data collection strategy combined with human preference capture and a joint objecting learning method to boost the performance of pre-trained dialogue models. The paper is very well-written, contains descriptions of the collected dataset alongwith  data analysis, as well as a learning strategy that leverages joint training via cross-entropy and contrastive learning (contrasting human preference dialog with model generated and random responses). The paper also conducts extensive human evaluation of the model responses alongwith ablation studies.

**Summary Of The Review:**

Overall the paper presents a very solid approach, dataset contribution and evaluation results for open-ended chatbot development in Chinese. The model fine-tuning method significantly improves the performance of the pre-trained models. The paper has more NLP results innovation than a learning representation innovation. The joint modeling strategy is used in many works today (NLL loss with a contrastive loss) so the technical novelty isn't the highlight of this very solid contribution.

---

> ### Author Response · Authors · 2022-11-19
> **Thank you! We have updated the paper to include the annotation interface with topic suggestions and the exploration of joint training on other datasets.**
>
> We thank the reviewer for their comments and suggestions! We have revised the paper to include the annotation interface with topic suggestions and the exploration of joint training on other datasets.
>
> > Authors mention they suggest topics in the guidelines but there isn't any AI assisted or explicit strategy to encourage users to discuss diverse topics.
>
> Thanks for catching this. In this revision, we have included the annotation interface in crafting the dialogue opening (Figure 5 of the Appendix). In the beginning, we will provide several two-level topics for inspiration (these topics are selected from the class hierarchy of Baidu Tieba [1]). If clicking the bottom button, it will display another set of topic suggestions. The annotator can craft the dialogue opening from scratch or based on any topic of interest.
>
> > The model training does not explicitly seem to leverage the revisions which would have been interesting to model.
>
> Currently, Diamante only leverages explicit human response (selected, revised, or rewritten by annotators) and implicit human preference. We include the final human response and intermediate model-generated candidates in the released dataset.
>
> Thanks for the suggestion. We will explore the explicit modeling of the revisions in our following work. It might be interesting to simultaneously encourage the generation of revised responses and suppress the generation of pre-revised responses.
>
> > Similar analysis on another dataset (maybe another language) would convince the readers much better on the strength of the model fine-tuning.
>
> Thanks for the suggestion. In this revision, we also explore applying the joint training framework to other conventional dialogue datasets without candidate responses, with DuSinc [2] taken as an example. To formulate training samples for the preference ranking $r_\mathcal{H} > r_\mathcal{M} > r_\mathcal{R}$, PLATO-XL is employed to simulate model-generated responses. The results (bottom bar in Figure 8 of the Appendix, also summarized in the following) verify the effectiveness and generality of the joint training paradigm.
>
> |          | w/ Joint Training  Win | Tie | w/o Joint Training  Win |
> | :--------| :---: | :---: | :---: |
> | DuSinc | 39\% | 52\% | 9\% |
>
> Notably, the data collection process in Diamante guarantees that the final response is preferred over model-generated candidates. While in DuSinc, some simulated model-generated candidates might not be inferior to human-produced responses, introducing some noise to the joint training. Despite this, the application of joint training boosts the performance of response generation in DuSinc.
>
> [1] https://tieba.baidu.com/f/index/forumclass
>
> [2] Han Zhou, et al., Link the world: Improving open-domain conversation with dynamic spatiotemporal-aware knowledge. arXiv:2206.14000, 2022.

---

### Official Review · Reviewer_tDkV · 2022-10-28

**Confidence:** 4
**Correctness:** 3
**Technical Novelty And Significance:** 2
**Empirical Novelty And Significance:** 3
**Recommendation:** 6

**Clarity, Quality, Novelty And Reproducibility:**

The originality of the work is from the dataset collected and the new training paradigm proposed to model response generation and model human preference.

**Strength And Weaknesses:**

Strengths and Weakness:
1. Collection of a new crowd sourced dataset for open domain conversational agent in Chinese Language.  How's is the quality of the dataset measure? What are steps taken to avoid bias and safety issues from being part of the dataset?
2. The joint training-generation framework is interesting in terms of modeling response generation. What is hypothesis behind having r_h > r_m > r_r?
3. Results show promising gains for the proposed approach on human evaluation both static and self chat evaluation.
4. Looking at the results from Table 6, it seems like most of the gains come from the dataset. The gains from joint training seems pretty minimal compared to Plato-XL.


**Summary Of The Paper:**

In this work, the authors propose a new framework called Diamante that is aimed at improving the performance of open domain chatbots by incorporating human feedback (both explicit and implicit preferences).  The proposed framework has a generation-evaluation training paradigm that is aimed to optimized response generation and preference estimation together.

Contributions:
1. The authors collect a new dataset for open domain chitchat conversation in Chinese and is called the Diamante dataset
2. A new joint generation and evaluation framework for optimizing generation and human preference estimation together.
3. Results from automated and human evaluation show that the existing approaches trained with the new training strategy achieves significant gains on metrics.

**Summary Of The Review:**

In this work, authors propose a new dataset for open domain conversations in Chinese and a new training paradigm that achieves gains in performance through human evaluation.

---

> ### Author Response · Authors · 2022-11-19
> **Thank you! We have updated the paper to include more analysis on joint training.**
>
> We thank the reviewer for their feedback and questions! We have revised the paper to include more ablation studies on joint training and explore applying joint training to other conventional datasets.
>
> > How's is the quality of the dataset measure? What are steps taken to avoid bias and safety issues from being part of the dataset?
>
> To ensure the annotation quality of the Diamante dataset, we designed and followed a rigorous quality control process, which is briefly summarized in Appendix B.2. During this process, several steps could alleviate or avoid potential bias or safety issues. 1) We engaged with a vendor company (whose core business is professional data annotation) to recruit experienced annotators. In total, 147 annotators passed the admission tests and participated in the dataset collection. 2) We provided detailed instructions in the guidelines and asked annotators to conduct conversations politely. We also provide good and bad examples in the guideline file to better illustrate the desirable quality. 3) After annotation, we would remove the conversation whenever one expert deems it ineligible, including coherence, safety, bias, etc. We have detailed definitions of safety according to Chinese cultural and social norms, including harmful speech (e.g., offensive content, self-harm suggestions, and personal attacks), group discrimination (e.g., region, gender, disability, and religion), misleading information (e.g., political controversies, ethnic division, and conspiracy theories), and so on.
>
> > The joint training framework is interesting in terms of modeling response generation. What is hypothesis behind having $r_h > r_m > r_r$?
>
> Thanks for raising this point. In Diamante's data collection, annotators are provided with model-generated candidates and asked to select, revise or rewrite the candidate to produce an appropriate response. If there exists one appropriate response in the candidates, the annotator can select it directly; otherwise, the annotator needs to revise or rewrite one appropriate response. With such a design of data collection, there exists implicit human preference: given the dialogue context $c$, the final response $r_\mathcal{H}$ is preferred by human annotators as compared to a model-generated candidate $r_\mathcal{M}$ (displayed during annotation). For a randomly selected response $r_\mathcal{R}$, it is not even coherent with the dialogue context in most cases. As such, we can have the following preference ranking in Diamante $r_\mathcal{H} > r_\mathcal{M} > r_\mathcal{R}$. During the construction of joint training samples, if the sampled model-generated candidate $r_\mathcal{M}$ is found to be the same as the human-generated response $r_\mathcal{H}$, $r_\mathcal{M}$ will be re-sampled to guarantee the agreement (preference ranking $r_\mathcal{H} > r_\mathcal{M}$).
>
> In this revision, we also explore applying the joint training framework to other conventional dialogue datasets without candidate responses, with DuSinc [1] taken as an example. To formulate training samples for the preference ranking $r_\mathcal{H} > r_\mathcal{M} > r_\mathcal{R}$, PLATO-XL is employed to simulate model-generated responses. The results (bottom bar in Figure 8 of the Appendix) verify the effectiveness and generality of the joint training paradigm.
>
> > Looking at the results from Table 6, it seems like most of the gains come from the dataset. The gains from joint training seems pretty minimal compared to Plato-XL.
>
> In this revised version, we have added the significance test for the human evaluation results (independent two-sample $t$-test, $p < 0.005$). In this table, Diamante has achieved significant improvements across the evaluation metrics except for coherence (the coherence is already 1.948, approaching the upper bound of 2). In addition to the listwise evaluation, we further carry out the pairwise comparison between models with and without joint training. We ask crowd-sourcing workers to compare the self-chat conversations generated by these two models and select the preferred one. The evaluation results (upper bar in Figure 8 of the Appendix) exhibit that the joint training paradigm is crucial in boosting the open-domain chatbot.
>
> [1] Han Zhou, et al., Link the world: Improving open-domain conversation with dynamic spatiotemporal-aware knowledge. arXiv:2206.14000, 2022.

---

### Official Review · Reviewer_fd1Y · 2022-11-02

**Confidence:** 3
**Correctness:** 2
**Technical Novelty And Significance:** 2
**Empirical Novelty And Significance:** 3
**Recommendation:** 5

**Clarity, Quality, Novelty And Reproducibility:**

This paper is well-written and easy to follow. The quality of the script is good, although there are still several typos such as "18% of the utterances" rather than "18% utterances". However, more examples and details of the human evaluations should be provided to support the claims of the authors. Moreover, the novelty is incremental since the proposed method is the composition of existing models, especially when compared with the similar previous effort LaMDA. It seems that the authors have released the dataset and published their code, which should be less challenging to reproduce their results.

**Strength And Weaknesses:**

Strength:

- Well written and easy to follow

- It is the first in Chinese to consider human preference in pretrained dialogue models based on human evaluations

- Experimental results show a significant improvement over existing methods in all metrics

Weaknesses:

- Limited novelty: It seems that the major difference between Diamante and LaMDA which also focuses on the alignment of human preference is the language. Although the authors claim that their joint-learning objective can be more effective than sequential learning in LaMDA, there are no experiments to support it. Authors should further clarify their contributions which currently are incremental as the combination of LaMDA and Chinese dialogue models such as PLATO-XL

- Insufficient details: Since the alignment of human preference relies on the quality of human evaluations, it is fundamental to demonstrate essential details such as the demographics of the crowd workers or the instructions for them to annotate, which is included in LaMDA. Authors are suggested to provide these details to address possible concerns about the quality of the human annotations. Moreover, if the annotations are implicit, it should be difficult for methods to distinguish the difference between safe and unsafe utterances. Thus authors should also provide a detailed description of their dataset, including the average turns of the conversation, and how and why these utterances are selected, revised, or rewritten from machine-generated utterances.

- Missing Examples: Authors only present the examples of Diamante, which can not demonstrate the improvement without the comparison of the examples of baselines including PLATO-XL and common commercial chatbots. Besides, the topics of the examples only cover insensitive topics such as food and people. Authors should show more examples of their methods and baselines for generating responses to sensitive topics such as Society or Culture. It is unclear if Diamente can yield more safety responses like LaMDA, especially on these sensitive topics.

**Summary Of The Paper:**

In this paper, authors propose a new framework named Diamante to improve the Chinese open-domain pre-trained dialogue models with human annotations, since these models have difficulty in generating engaging utterances during the conversation. Diamante first builds a Chinese dataset by requiring human annotators to choose or amend the model-generated responses. Based on the dataset, it designs a joint-training process that considers both the generation loss and the human preference estimation loss. Experimental results demonstrate the effectiveness of Diamante compared with existing Chinese open-domain pre-trained dialogue methods.

**Summary Of The Review:**

The paper is well written and focuses on an important issue in Chinese dialogue models that there are no previous efforts to consider human preferences. However, the authors should better clarify their difference to LaMDA and the benefits of their proposed joint training based on implicit annotations. To support their claims, more details of the human evaluations and dialogue examples should also be presented. Based on the current version, I would not recommend accepting this paper.

---

> ### Author Response · Authors · 2022-11-19
> **Thank you! We have updated the paper to include more annotation details, qualitative examples of compared approaches, and safety analysis under adversarial attacks.**
>
> We appreciate the comments and suggestions from the reviewer! We have revised the paper to include more details of dataset annotation, qualitative examples of compared approaches, and safety analysis under adversarial attacks.
>
> > Although the authors claim that their joint-learning objective can be more effective than sequential learning in LaMDA, there are no experiments to support it.
>
> As discussed in the last paragraph of Section 3, there are three main differences between Diamante and LaMDA: simultaneous joint optimization vs. sequential learning, implicit overall human preference vs. additional fine-grained evaluation annotation, and ranked preference estimation vs. binary discrimination. Compared with LaMDA, Diamante leverages implicit human preference as the overall evaluation and gets rid of additional annotations. Moreover, in preliminary experiments, we implemented one baseline with binary discrimination and sequential learning. Here is the comparison result on the validation set (PPL reflects the performance on response generation; MAP, MRR, and P@1 reflect the performance on response ranking):
>
> |          | PPL | MAP | MRR | P@1 |
> | :--------| :---: | :---: | :---: | :---: |
> | Baseline | 9.409 | 0.493 | 0.485 | 0.273 |
> | Diamante | 9.453 | 0.699 | 0.693 | 0.544 |
>
> In the baseline experiments, we have tried careful hyper-parameter tuning and separated optimizing parameters to alleviate the issue of catastrophic forgetting as much as possible. Given the training difficulty and relatively inferior performance, we stick to the joint optimization with ranked preference estimation in our experiments.
>
> > It is fundamental to demonstrate essential details such as the demographics of the crowd workers or the instructions for them to annotate.
>
> Thanks for raising this point. We engaged with a vendor company (whose core business is professional data annotation) to recruit experienced annotators. In total, 147 annotators participated in the dataset collection. These annotators have high school degrees or above. Due to privacy regulations, we could not get more detailed information on age, gender, or ethnicity from the vendor company.
>
> The annotation instructions are discussed in Section 2.1, and more details are provided in Appendix B. In this revision, we have also included the original guideline and admission test files in the supplementary materials (these files are in Chinese).
>
> > Authors only present the examples of Diamante, which can not demonstrate the improvement without the comparison of the examples of baselines including PLATO-XL and common commercial chatbots.
>
> Thanks for the suggestion. Due to the page limit, we only present the examples of Diamante in the main text. In this revision, we have included the examples of PLATO-XL and XiaoIce (as the representative of commercial chatbots) in the Appendix. These examples are under the same starting utterances as Diamante in Figures 3 and 4.
>
> > It is unclear if Diamante can yield more safety responses like LaMDA, especially on these sensitive topics.
>
> Thanks for pointing this out. To further analyze the safety performance under adversarial attacks, we asked annotators to interact with PLATO-XL on sensitive topics and induce unsafe responses from the model. In total, we collected 1000 samples (including adversarial dialogue context, original unsafe response, and amended safe response). The collection settings and experimental results are discussed in Appendix C.2. The evaluation results reveal that Diamante achieves a remarkable safety improvement, with 76\% of generated responses identified as safe. Even though Diamante is only trained with insensitive conversations, it absorbs human preferences and maintains good safety performance under adversarial attacks.

---

### Official Review · Reviewer_Q4VE · 2022-11-03

**Confidence:** 4
**Correctness:** 3
**Technical Novelty And Significance:** 2
**Empirical Novelty And Significance:** 3
**Recommendation:** 6

**Clarity, Quality, Novelty And Reproducibility:**

The paper is well-written and easy to understand. Some methods used in this work are derived from existing work.

**Strength And Weaknesses:**

### Strengths


Including human preferences and feedback in generation models is an important research direction. The paper takes meaningful steps to achieve this in a non-English setting. The methodology for data collection is sensible, and the scale of the dataset is sufficiently large to be of interest to the broader community.


### Weaknesses

1. In the preference estimation loss (Equation 2), the authors always assume that human-generated responses are better. However, it is mentioned in Table 1 that 18% of the responses were selected as being already good. Isn’t there a contradiction here? Essentially, Equation 2 will sometimes force the model to learn *not* to select an otherwise perfectly valid response?

2. **Evaluation**

2.1: This is probably a clarification: was there any overlap in the set of human annotators the same for data creation and evaluation (the paper mentions “in-house data specialists”)? If yes, there is a risk that the method overfits not to human preferences but to biases of a select set of reviewers. For example, consider 5 annotators: A, B, C, D. A and B like cheerful responses, and C and D like brief, serious responses.

Suppose the responses from the baseline are either cheerful or serious with a 50% probability.
If we collect data using annotators A, B and also make them evaluate, the baseline performance will be ~ 50%, and your method will achieve a score of nearly 100%. The other extreme follows from using C, D as evaluators. Thus, the right way to go about it is to randomly select the groups for evaluation and data generation.


2.2. “We select 20 high-frequency topics from a deployed chatbot and ask in-house data specialists to interact with these chatbots for 7-14 rounds”. How much data was used for evaluation?

2.3 The annotator agreement is moderate (closer to the boundary of low and moderate), indicating that the method may generate better responses simply because of more fine-tuning.

3. **Engagement vs. utility:** The key assumption in this work is that engagement is proportional to human preferences. Is that necessarily true? For example, a chatbot that produces brief/terse responses may not be preferred for conversation in general but might be okay for a chatbot as long as it’s useful.

Other notes/good to have:

a) engaginess should probably be changed to “engagement.”

b) For human evaluation results in Table 2, significance tests are missing.


**Summary Of The Paper:**


This work aims to train chatbots that generate responses aligned with human preferences. The training dataset is generated with human annotation. Specifically, the annotators start a conversation with a chatbot on a certain topic. At each turn, the annotators can provide feedback on the generated response. The feedback can come in the form of revisions to the generated response or a complete rewrite. After correction, the dialogue is continued, and the “response → feedback” steps are repeated.

During fine-tuning, the corrected responses are used in the standard perplexity loss. Additionally, a “preference estimation” loss is included to encourage the model to rank the corrected responses over the original model response.

Humans rate the responses generated by the proposed model to be more coherent, informative, and engaging.


**Summary Of The Review:**

Making text generation methods sensitive to feedback is an important research direction, and this paper makes useful contributions. My current recommendation is a weak accept, but I’ll change my scores after the authors’ feedback.

---

> ### Author Response · Authors · 2022-11-19
> **Thank you! We have updated our paper to reflect the reviewer's recommendations.**
>
> We thank the reviewer for their feedback and questions! We have revised the paper to include clarification on training sample construction and human evaluation results with significance tests.
>
> > In the preference estimation loss (Equation 2), the authors always assume that human-generated responses are better. However, it is mentioned in Table 1 that 18\% of the responses were selected as being already good. Isn’t there a contradiction here?
>
> Thanks for pointing this out. During the construction of joint training samples, if the sampled model-generated candidate $r_\mathcal{M}\in \mathcal{R}_\mathcal{M}$ is found to be the same as the human-generated response $r_\mathcal{H}$, $r_\mathcal{M}$ will be re-sampled to guarantee the agreement (preference ranking $r_\mathcal{H} > r_\mathcal{M}$).
>
> > Was there any overlap in the set of human annotators the same for data creation and evaluation?
>
> There is no overlap between dataset annotators and dialogue evaluators. We engaged with a vendor company to recruit experienced annotators for dataset creation. We employed another group of workers from a crowd-sourcing platform for dialogue evaluation.
>
> > ``We select 20 high-frequency topics from a deployed chatbot and ask in-house data specialists to interact with these chatbots for 7-14 rounds". How much data was used for evaluation?
>
> For comparison to common commercial chatbots (Table 4), we asked in-house data specialists to interact with these chatbots for 7-14 rounds, such that the conversations can continue and end naturally. To make the evaluation volumes equal, we truncate all chatbots' dialogues at the 7th round.
>
> The data amount used for evaluation in Table 4 is summarized as follows.
> - 20 topics * 5 chatbots * 7 utterances (utterance-level evaluation), in total 700 utterances, each distributed to 3 crowd-sourcing workers and evaluated on 3 utterance-level metrics;
> - 20 topics * 5 chatbots (dialogue-level evaluation), in total 100 dialogues, each distributed to 3 crowd-sourcing workers and evaluated on the dialogue-level metric.
>
> > The annotator agreement is moderate (closer to the boundary of low and moderate) [...] For human evaluation results in Table 2, significance tests are missing.
>
> ``The Fleiss' kappa score for the static evaluation, self-chat evaluation, and human-bot chat evaluation is 0.433, 0.468, and 0.424, respectively." We agree these Fleiss' kappa scores are not very high, just comparable or slightly higher to other open-domain chitchat evaluations in Chinese (PLATO-2, 0.466) or English (Meena, 0.36). The detailed scoring criteria provided to crowd-sourcing workers are included in Appendix A, hoping to shed some light to our community on formulating evaluation standards and facilitating annotator agreements in the future.
>
> Thanks for the suggestion on the significance test. The statistically significant improvements (independent two-sample $t$-test, $p < 0.005$) are now indicated in the human evaluation results (Table 2, 3, 4, 5, 6). Diamante almost achieves significant improvements across all these evaluations.
>
> > Engagement vs. utility: The key assumption in this work is that engagement is proportional to human preferences. Is that necessarily true? For example, a chatbot that produces brief/terse responses may not be preferred for conversation in general but might be okay for a chatbot as long as it’s useful.
>
> Thanks for raising this point. Engagement is not always proportional to human preference across different dialogue systems. The conversational AI typically includes three categories: open-domain chitchat, knowledge-grounded conversation, and task-oriented dialogue. In this work, we focus on open-domain chitchat, where overall engagement is the main objective and proportional to human preference. While in task-oriented dialogue, task completion instead of engagement will become the main objective. Since Diamante is a general framework to boost dialogue systems towards human preference, we have plans to explore Diamante in other kinds of conversations soon.

---

### Decision · Program_Chairs · 2023-01-20

**Decision:**

Reject

**Justification For Why Not Higher Score:**

A lot of concerns are raised by reviewers, mainly including: Limited novelty: the idea is not new, and has been explored in a lot of previous work.  The evaluation is not very convincing in terms of metrics, dataset and baselines.  Some details are missing and the analysis is not sufficient.

**Justification For Why Not Lower Score:**

NA

**Metareview: Summary, Strengths And Weaknesses:**

Summary:

The authors propose a new framework to improving the performance of open domain Chinese chatbots by incorporating human feedback (both explicit and implicit preferences). A generate-then-rerank paradigm is used to first generate multiple candidates and then select a preferred one. The training dataset is generated with human annotation. A joint generation and evaluation framework is designed that considers both the generation loss and the human preference estimation loss. Results from automated and human evaluation show that the existing approaches trained with the new training strategy achieves signi cant gains on metrics of coherence, informativeness and engagement.

Strengths:

Including human preferences and feedback in generation models is an important research direction.
Collection of a new crowd sourced dataset for open domain conversational agent in Chinese Language. The methodology for data collection is sensible, and the scale of the dataset is sufficiently large to be of interest to the broader community.
The joint training-generation framework is interesting, combining mask learning with contrastive learning using implicit human preferences.
Experimental results show a significant improvement over existing methods in human evaluation.
Ablation studies are conducted.
The paper is well-written and easy to follow.

Weaknesses:

A lot of concerns are raised by reviewers, mainly including: Limited novelty: the idea is not new, and has been explored in a lot of previous work.  The evaluation is not very convincing in terms of metrics, dataset and baselines.  Some details are missing and the analysis is not sufficient.



**Summary Of Ac-Reviewer Meeting:**

NA